# Spatial maps of prostate cancer transcriptomes reveal an unexplored landscape of heterogeneity

Emelie Berglund[1], Jonas Maaskola[1], Niklas Schultz[2], Stefanie Friedrich[3], Maja Marklund[1], Joseph Bergenstråhle[1], Firas Tarish[2], Anna Tanoglidi[4], Sanja Vickovic [1], Ludvig Larsson[1], Fredrik Salmén[1], Christoph Ogris[3], Karolina Wallenborg[2], Jens Lagergren[5], Patrik Ståhl[1], Erik Sonnhammer[3], Thomas Helleday[2] & Joakim Lundeberg [1]

Intra-tumor heterogeneity is one of the biggest challenges in cancer treatment today. Here we investigate tissue-wide gene expression heterogeneity throughout a multifocal prostate cancer using the spatial transcriptomics (ST) technology. Utilizing a novel approach for deconvolution, we analyze the transcriptomes of nearly 6750 tissue regions and extract distinct expression profiles for the different tissue components, such as stroma, normal and PIN glands, immune cells and cancer. We distinguish healthy and diseased areas and thereby provide insight into gene expression changes during the progression of prostate cancer. Compared to pathologist annotations, we delineate the extent of cancer foci more accurately, interestingly without link to histological changes. We identify gene expression gradients in stroma adjacent to tumor regions that allow for re-stratification of the tumor micro-environment. The establishment of these profiles is the first step towards an unbiased view of prostate cancer and can serve as a dictionary for future studies.

[1] Department of Gene Technology, School of Engineering Sciences in Chemistry, Biotechnology and Health, Royal Institute of Technology (KTH), Science for Life Laboratory, Tomtebodavägen 23, Solna 17165, Sweden. [2] Department of Oncology-Pathology, Karolinska Institutet (KI), Science for Life Laboratory, Tomtebodavägen 23, Solna 17165, Sweden. [3] Department of Biochemistry and Biophysics, Stockholm University, Science for Life Laboratory, Tomtebodavägen 23, Solna 17165, Sweden. [4] Department of Clinical Pathology, University Uppsala Hospital, Rudbecklaboratoriet, Uppsala 751 85, Sweden. [5] Department of Computational Biology, School of Computer Science and Communication, Royal Institute of Technology (KTH), Science for Life Laboratory, Tomtebodavägen 23, Solna 17165, Sweden. These authors contributed equally: Emelie Berglund, Jonas Maaskola, Niklas Schultz, Stefanie Friedrich Correspondence and requests for materials should be addressed to T.H. (email: thomas.helleday@scilifelab.se) or to J.L. (email: joakim.lundeberg@scilifelab.se)

Prostate cancer (PCa) is the most common type of cancer among men worldwide[1], causes annually over 250,000 deaths, and advances via clonal evolution[2]. Frequently, multiple competing clones with independent tumor origins exist within the primary tumor[3–5]. Moreover, acquired somatic events by the clones increase the probability for metastasis[6]. Consequently, PCa contains substantial intratumoral heterogeneity with genetic alterations present both in the original tumor and in distant metastases[7,8]. Subclonal diversity can be analyzed with DNA-sequencing of a bulk tumor sample[7,9] or more precisely by using laser capture technology[3]. While genetic changes are important to track cancer heterogeneity and clonal evolution they may also be of clinical relevance. This is exemplified by the high proportion of castration-resistant PCa cases being DNA repair deficient[10]. In these cases frequent mutations in BRCA2 and ATM genes are linked to successful treatment with PARP inhibitors[11]. Furthermore the tumor microenvironment in the form of reactive stroma plays a functional role during inflammation and in neoplastic transformation[12].

By using single-cell RNA-Seq (scRNA-Seq), intra-tumor gene expression heterogeneity has been documented at the level of individual cells[13–15] and advancements in droplet microfluidics and barcoding have made it possible to analyze thousands of cells[16]. The lack of spatial information for scRNA-Seq data can, to a certain extent, be circumvented by computational inference[17,18]. Current in situ sequencing techniques have, until recently, been limited to measuring small numbers of genes[19,20] and the spatial dimensions of entire transcriptomes remain unexplored in PCa, along with the tumor microenvironment.

Pathological severity of prostate adenocarcinoma, despite progress with molecular markers and MRI, is generally scored according to the Gleason grading (Gs) system, which uses histological data only, frequently complemented with PSA measurement in blood and tumor staging[21]. However, this classification method has limitations and new alternatives have been proposed[22].

Here we investigate for the first time a multifocal PCa simultaneously at tissue- and transcriptome-wide scale using the recently introduced Spatial Transcriptomics (ST) method[23], which allows for quantification of the mRNA population in the spatial context of intact tissue. We use a novel computational procedure to elicit spatial, transcriptome-wide expression patterns enabling deconvolution of molecular events in cancer and associated microenvironment.

## Results

**Measuring spatial gene expression in prostate cancer tissue sections.** The study design involves twelve spatially separated biopsies taken from a cancerous prostate after radical prostatectomy (Gs 3 + 4, pT3b, PSA = 7.1)(Fig. 1a). We measured spatial gene expression throughout twelve tissue sections using the ST methodology (Fig. 1b, Supplementary Fig. 1a). Supplementary Table 1 contains a summary of the data evaluation. Overall, 5 910 tissue regions within the 12 sections were analyzed.

**Transcriptome heterogeneity in the spatial vicinity of a cancer.** We initially analyzed one tissue section containing a tumor (sample 1.2, Gs 3 + 3, Supplementary Fig. 1b). We used a novel factor analysis method (Supplementary methods) to infer activity maps (Fig. 2a, Supplementary Fig. 2a, Supplementary Data 1) and expression profiles (Supplementary Fig. 3, Supplementary Data 1). The factors' activity maps generally exhibit spatial patterns that closely mirror histologically identifiable structures, such as normal glands and stroma (Fig. 2a, Supplementary Fig. 2a). Others overlap with regions annotated as cancerous or prostatic intraepithelial neoplasia (PIN) (Fig. 2a, b). Remaining factors are annotated by combined analysis of listed top genes, histology and calculated proportion of stroma, epithelium and lumen (Supplementary Fig. 2a, Supplementary Table 2). Notably, the "cancer" factor is active in a region that encompasses the annotated cancer region. Hierarchical clustering of the ST read count data (Supplementary Fig. 4a) is consistent with the factor activity patterns (Fig. 2a). In addition, principal component analysis (PCA) confirms clear separation between regions (Fig. 2d). The gene expression profiles (Supplementary Fig. 3) generally reflect the expression phenotype of the prostate as tissue of origin or the functional requirements of the respective tissue components. For example, KLK3, KLK2, MSMB, and ACPP are among the highest-expressed genes in many of the factors. In the "stromal" factor the top-expressed genes have functions related to cytoskeleton, smooth muscle and cell-adhesion. Known PCa-related genes

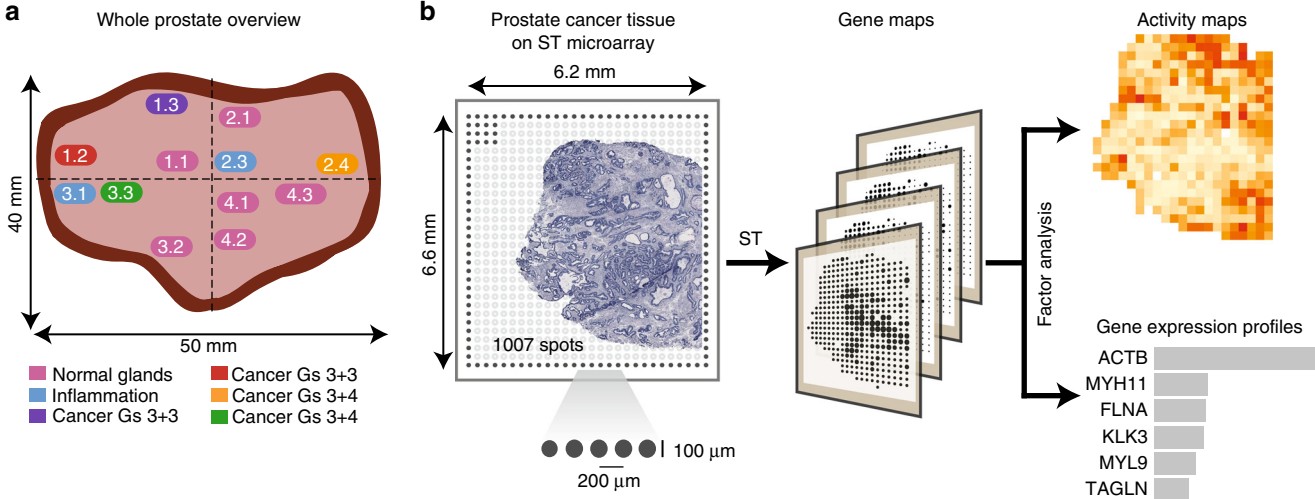

**Fig. 1** Study design for spatial transcriptomics (ST) in prostate cancer. **a** Location of sections used in this study and annotations made by a pathologist. The sections are colored according to annotation. Scale bars indicate size of the prostate. **b** Spatial microarrays have 1007 spatially barcoded spots of 100 μm diameter and 200 μm center-to-center distance. Spots denoted by filled circles are used for orientation, and lack spatial barcodes. The ST procedure yields matrices with read counts for every gene in every spot, which are then decomposed by factor analysis resulting in a set of factors ("cell types"), each comprising one activity map and one expression profile

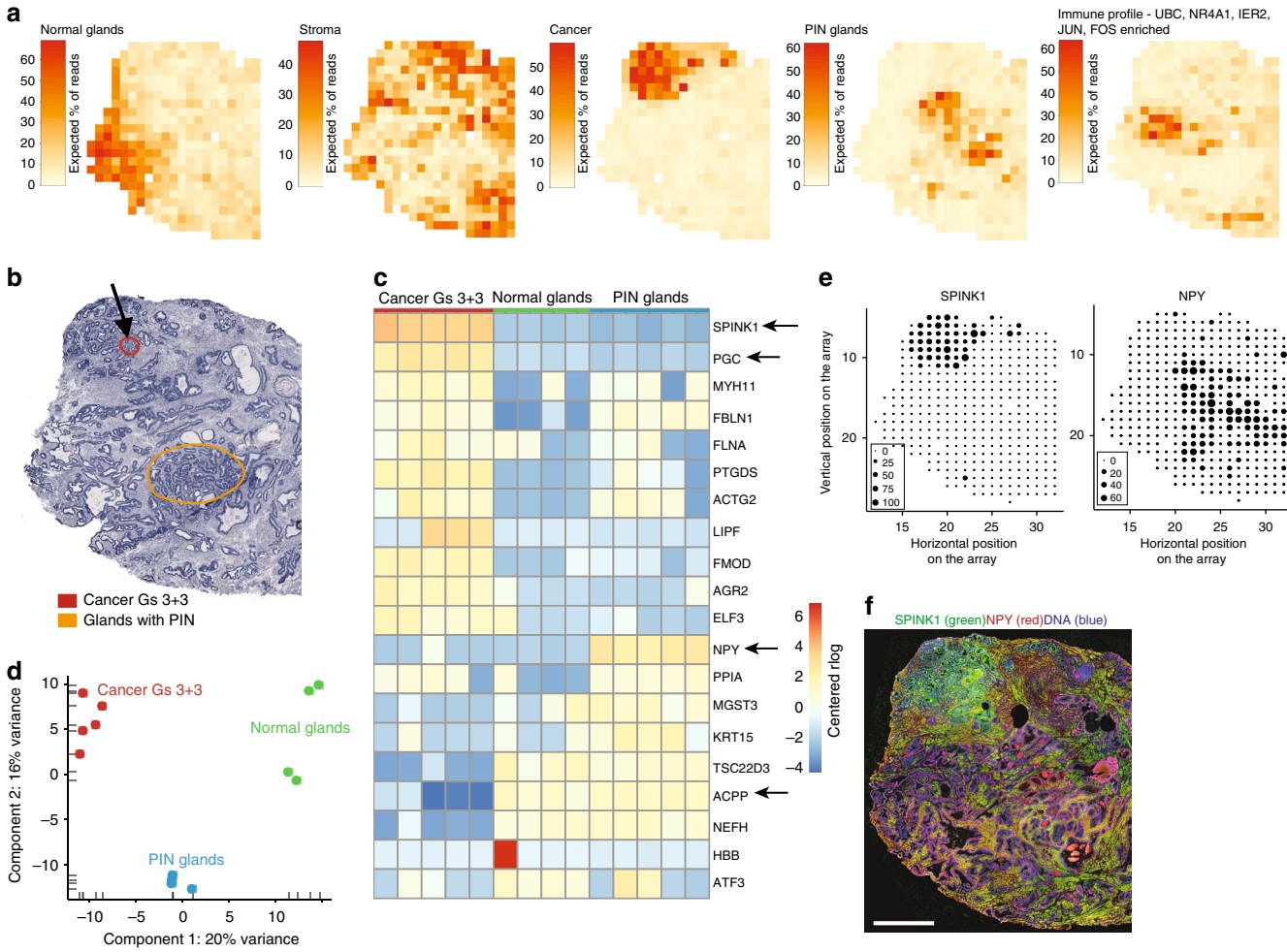

**Fig. 2** Spatial gene expression heterogeneity within the 1.2 cancer tissue sample. **a** Factor activity maps for selected factors corresponding to epithelial, stromal, cancerous, PIN, or inflamed regions. Remaining factors' activity maps in Supplementary Figure 2 and Supplementary Data 1. **b** Annotated brightfield image of H&E-stained tissue section. **c** Heatmap of the 20 most variable genes between cancer, PIN and normal gland regions, using spot sets from Supplementary Fig. 4b. Centered rlog: difference of rlog (variance-stabilized transform of ST expression data) and gene-wise mean rlog. Arrows highlight genes of interest validated by immunohistochemistry (IHC). **d** First two principal components of spot sets from **c** separate cancer, PIN and normal regions. **e** Array dot plots for SPINK1 and NPY. Circle size in array dot plots indicates normalized ST counts. **f** IHC staining for SPINK1 and NPY of an adjacent section on the ST array. Nuclei are stained with DAPI (blue). Scale bar indicates 1 mm

(*SPON2, TFF3, SPINK1*) are among the highest-expressed genes in the "cancer" factor.

In order to identify interactions between the factors, we performed hierarchical clustering (Supplementary Fig. 2b) of all ten factors. The clustering yielded three main groups. Among the ten factors, inflammation, PIN and cancer, could delineate one of the groups, while being seperated from normal glands. This further indicate inflammation as a critical component of tumor initiation and progression. As expected, factors containing mainly stroma cells cluster together. Factor 1 contained high levels of *MSMB* and annotated as "normal glands signature". *MSMB* is known to be downregulated in prostate cancer[24].

The gene expression within each region (normal, cancer, PIN) obtained from the preceding factor analysis was investigated to identify region-specific markers (Supplementary Fig. 4b). In the cancer region we for instance observe enrichment of *SPINK1* and *PGC*, and depletion of *ACPP* (Fig. 2c, Supplementary Data 2). Another noteworthy observation is elevated levels of *NPY* in the PIN region (Fig. 2c). To validate these findings we investigated the concordance between gene expression and staining of the corresponding proteins within the tissue. Immunostaining of *SPINK1* was coincident with the defined cancer region and

immunostaining of *NPY* was mostly localized to that of the PIN region (Fig. 2e, f).

**Spatial expression patterns common to cancer tissue sections.** Next, we carried out a factor analysis for three tissue sections containing annotated cancer foci (Fig. 3). The resulting gene expression profiles (Supplementary Fig. 5, Supplementary Data 3) are similar to the preceding analysis. The activity maps (Fig. 3b, Supplementary Fig. 6a, Supplementary Data 3) again show patterns corresponding to annotated or histologically identifiable structures (Fig. 3a), but now group regions with similar phenotype across tissue sections, and for tissue section 1.2 are virtually identical to the preceding analysis. In particular, the "cancer" factor now includes in section 2.4 a region annotated as suspected cancer and also shows slight activity in Section 3.3 (Fig. 3b). *SPON2, TFF3*, and *SPINK1* are again among the highest-expressed genes for this factor (Supplementary Fig. 5). The gene expression profile of another factor reflects processes observed in reactive stroma ("reactive stroma", Supplementary Fig. 5) and surrounds that of the "cancer" factor in samples 2.4 and 3.3 (Fig. 3b). Although the pathologist marked section 3.3 with cancer and few PIN and normal glands, we detected

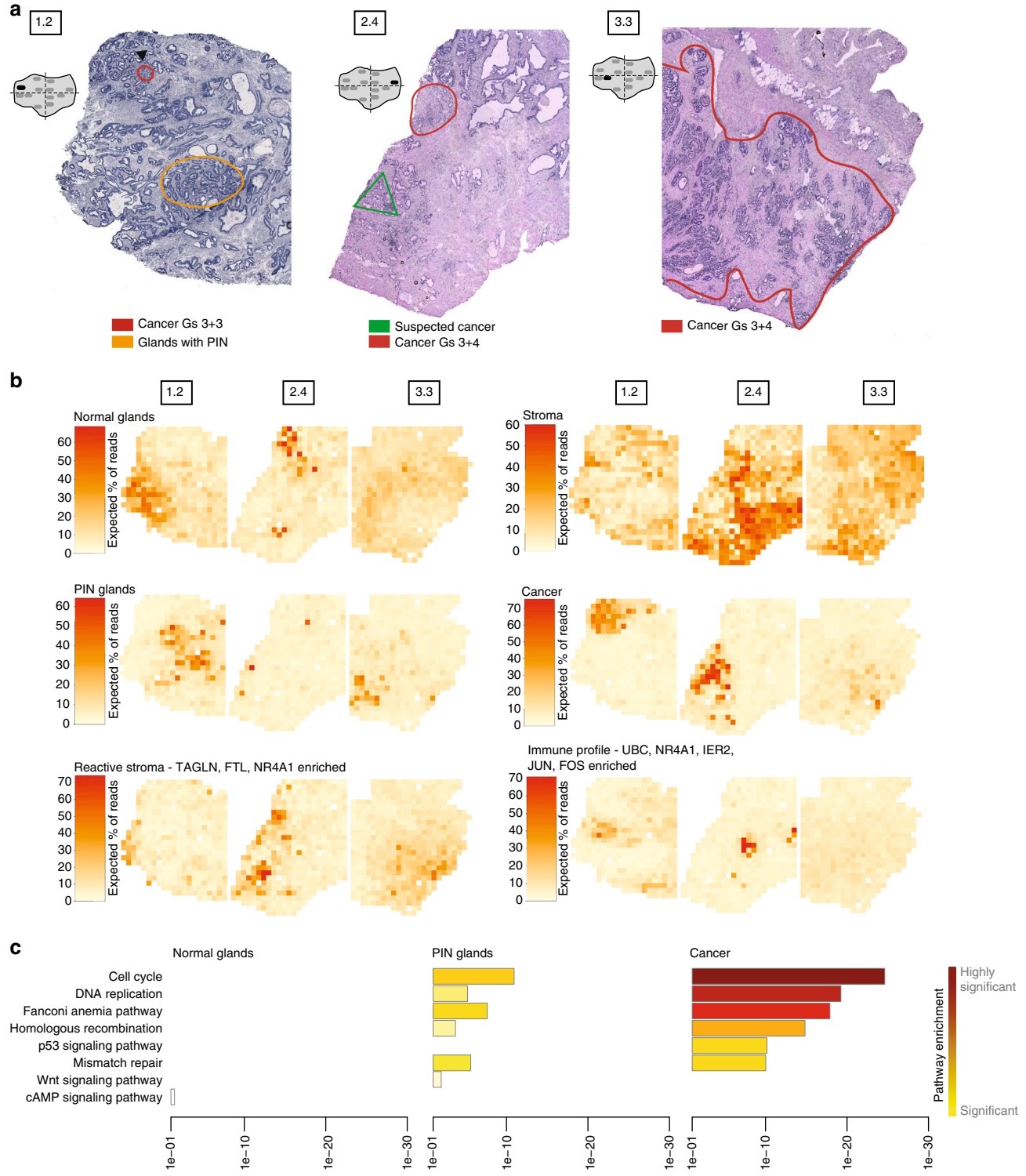

**Fig. 3** Histology and shared gene expression factors in three cancer tissue sections. **a** Annotated brightfield images of H&E-stained tissue sections. **b** Factor activity maps for selected factors corresponding to normal glands, stromal, PIN, cancerous, or inflamed regions based on a joint factor analysis of the three sections. Remaining factors' activity maps in Supplementary Figure 6 and Supplementary Data 3. **c** Pathways enriched in normal, cancer and PIN epithelium clearly differentiate healthy from diseased tissue. Signficant pathways were calculated using the Komolgorov–Smirnov normality test at the 0.05 alpha level

relatively large areas containing both normal and PIN glands within the cancer annotated region ("normal glands" and "PIN glands", Fig. 3b). This heterogeneity within the cancer area is supported by PCA and hierarchical clustering of spots taken from the annotated cancer area in sample 3.3 (Supplementary Fig. 7).

Notably, hierarchical clustering (Supplementary Fig. 6b) of the factors was similar to the preceding one (Supplementary Fig. 2b). Importantly, inflammation is once again linked to cancer and PIN, and a clear separation of normal glands (with and without *MSMB*) from stroma and malignant cells was observed.

Interestingly, we identified factor 1 to be a mix of stroma and normal epithelial cells expressing high levels of *RACK1* and *FTH1*. Studies has shown that *RACK1* is part of the tyrosine kinase signaling, facilitating transcriptional activity of the androgen receptor (AR)[25].

In addition to our genome-wide measurements, we investigated pathways in normal, PIN and cancer epithelium (Fig. 3c, Supplementary Fig. 8, Supplementary Data 4). Higher expression of cancer-related pathways was observed in cancer and PIN compared with normal epithelium. We observed increased expression of cell cycle, DNA replication, homologous recombination, Fanconi anemia and p53 signaling in the cancer and PIN areas. Notably, Wnt signaling was only detected in the PIN factor. This suggest that Wnt signaling plays an important role in the initiation and progression of PIN. Studies have shown association of Wnt signaling/β-catenin expression with tumor evolution[26]. All of the above mentioned pathways differentiate benign from malignant tissue and with this in mind, they could be particularly important in characterizing normal or pre-neoplastic tissue (PIN). It also opens up the possibility to develop anticancer treatments to target these pathways, for example cell cycle arrest that hinder progression of cells from the $G_1$ to S phase[27,28] or targeting DNA repair with PARP inhibitors[11]. Moreover, tumors with deficiencies in homologous recombination or Fanconi anemia proteins are known to respond to platinum-based chemotherapy[29].

Disappearance of basal cells from prostatic glands is one of the hallmarks of PCa, and the protein P63 is a well-established marker for basal cells[30]. We validated by IHC that presence or absence of P63-stained glands co-incident with pathological annotation and the ST factor analysis results (Supplementary Figs. 9–12). Notably, the absence of P63 staining in sample 1.2 is coincident with the "cancer" factor's activity, and extends further than indicated by the pathologist (Supplementary Fig. 10). In sample 2.4, no P63-stained glands are observed throughout an area that comprises the annotated and suspected cancer foci (Supplementary Fig. 11). While P63 staining is mostly absent in the large annotated cancer area of sample 3.3, some stray glands show P63 staining (Supplementary Fig. 12). Closer inspection reveals flat glands with P63 staining (Supplementary Figs. 9 and 12), consistent with PIN characteristics[31]. To confirm the ST expression data for several genes of interest, we performed immunohistochemistry (IHC) and found concordance between the spatial expression patterns of protein and mRNA (Supplementary Figs. 13–14).

To explore observation across tumors, we analyzed two more tumors from two patients (corresponding to an additional 750 tissue regions) and compared the transcriptional profiles of the tumor areas from the first patient (Supplementary Fig. 15a, Supplementary Data 5). We detect extensive tumor heterogeneity between patients as determined by the factor analysis. To further investigate, at the gene level, we could confirm some observations obtained from patient 1. Interestingly, we identified *NPY* (enriched in PIN regions), *SPON2* (enriched in cancer regions) and *NR4A1* (enriched in reactive stroma, see below) also in patient 2 and 3 (Supplementary Fig. 15b), while *EEF2, NEAT1*, and *TPT1* (established interaction with p53)[32] were uniquely expressed in patient 3 (Supplementary Fig. 16).

The transcriptome wide data for patient 2 looked quite distinct compared to patient 1 and 3. Among the highest genes, we found *SERPINA3* and *TPT1* (Supplementary Fig. 16).

**Gene expression differences in the center and the periphery of cancer.** We sought to in more detail investigate functional differences in gene expression between the center and the periphery

of the cancer, respectively and how the signals from the tumor stimulates the adjacent endothelium. Abnormal tissues adjacent to tumors were first described in 1953, also called the "field cancerization"[33]. Previous studies have suggested that breast cancer tissue close to cancer undergoes extracellular matrix remodeling, fibrosis, and an epithelial-to-mesenchymal transition (EMT)[34]. Other studies, focusing on prostate cancer investigated the gene expression differences among prostate cancer tissue, adjacent prostate cancer tissue and normal prostate tissue[35]. They found that the tumor vs. normal expression profile was more extensive than the tumor vs. adjacent normal profile. Also, tumor and adjacent tumor tissues emerged with higher response of inflammatory and immune than normal tissues, which was in agreement with previous report that inflammation was closely related to cancer. A recent study conducted on many different cancer types revealed that the adjacent tumor tissue represents an intermediate state between normal and cancer tissue[36]. They also uncovered activation of pro-inflammatory response in the adjacent tissue. However, no evaluation with spatial resolution of cancer and adjacent cancer tissue has been performed to date. We therefore aimed for discover differences between the cancer and the periphery of the cancer samples 1.2 (Gs 3 + 3), 2.4 (suspected cancer) and 3.3 (Gs 3 + 4) (Fig. 4 and Supplementary Figs. 17–18).

The pathways activated in the center of sample 1.2 (Fig. 4d) are mainly linked to altered cellular metabolism (Oxidative phosphorylation, Pentose phosphate pathway[37], Citrate cycle); metabolic alteration is a hallmark of cancer[38]. The activated TCA cycle pathway is essential for a neoplastic prostate cell to evolve into a malignant tumor cell[39] and its activations in the center suggests that the malignant center cells with high energy consumption are surrounded by pre-malignant cells. On the other hand, for sample 3.3 (Fig. 4h), higher levels of metabolism is seen in the periphery of the cancer. Furthermore, sample 3.3 expresses higher levels of Endocytosis, Phagosome and Lysosome in the central area. These pathways are known to be high in necrotic areas, in which they clear cell debris and dead cells[40]. Notably, we also observe enrichment of the HIF-1 signaling pathway. As a tumor grows, it causes abnormalities in tumor blood vessels, leaving region of the tumor with lower oxygen concentration compared to normal tissue[41]. The best understood mechanism of how cancer cells adapt to a hypoxic environment is through elevated levels of *HIF-1* and *HIF-2*[42].

Notably, the activated pathways in the periphery of 1.2 are mainly related to stress, inflammation (NF-kappa B signaling, Toll-like receptor signaling, NOD-like receptor signaling) (Fig. 4d). Such theory is supported by other studies[35,36]. We confirmed that the presence of immune pathways was due to presence of immune cells in the nearby tissue by asking a pathologist to annotate if inflammation exists (Supplementary Fig. 17a). We also found high levels of immune-related genes in the periphery (e.g., *IRF7, HLA-C* and *NFKBIA*) (Supplementary Fig. 17a).

Both tumors showed high levels of pathways linked to cell proliferation (MAPK signaling, ECM-receptor interaction, PI3K-Akt signaling) and cell motility (regulation of the actin cytoskeleton, focal adhesion), although they were found in different regions of 1.2 and 3.3 (either the center or periphery of the cancer) (Fig. 4d, h).

We see higher expression of some genes in the periphery compared to the center in all three cancer areas (Fig. 4 and Supplementary Figs. 17–18). For example, *TAGLN* (tumor suppressor)[43], *HLA* (linked to the immune system in humans)[44], *ACTG2* and *ACTB* (involved in cell motility)[45]. Genes that are higher expressed in the center in all three cancer areas are for example, *NUPR1, ASAH1 PDLIM5, KLK4* and *PSCA*. All are known to be highly expressed in PCa[46]. Additionally noteworthy

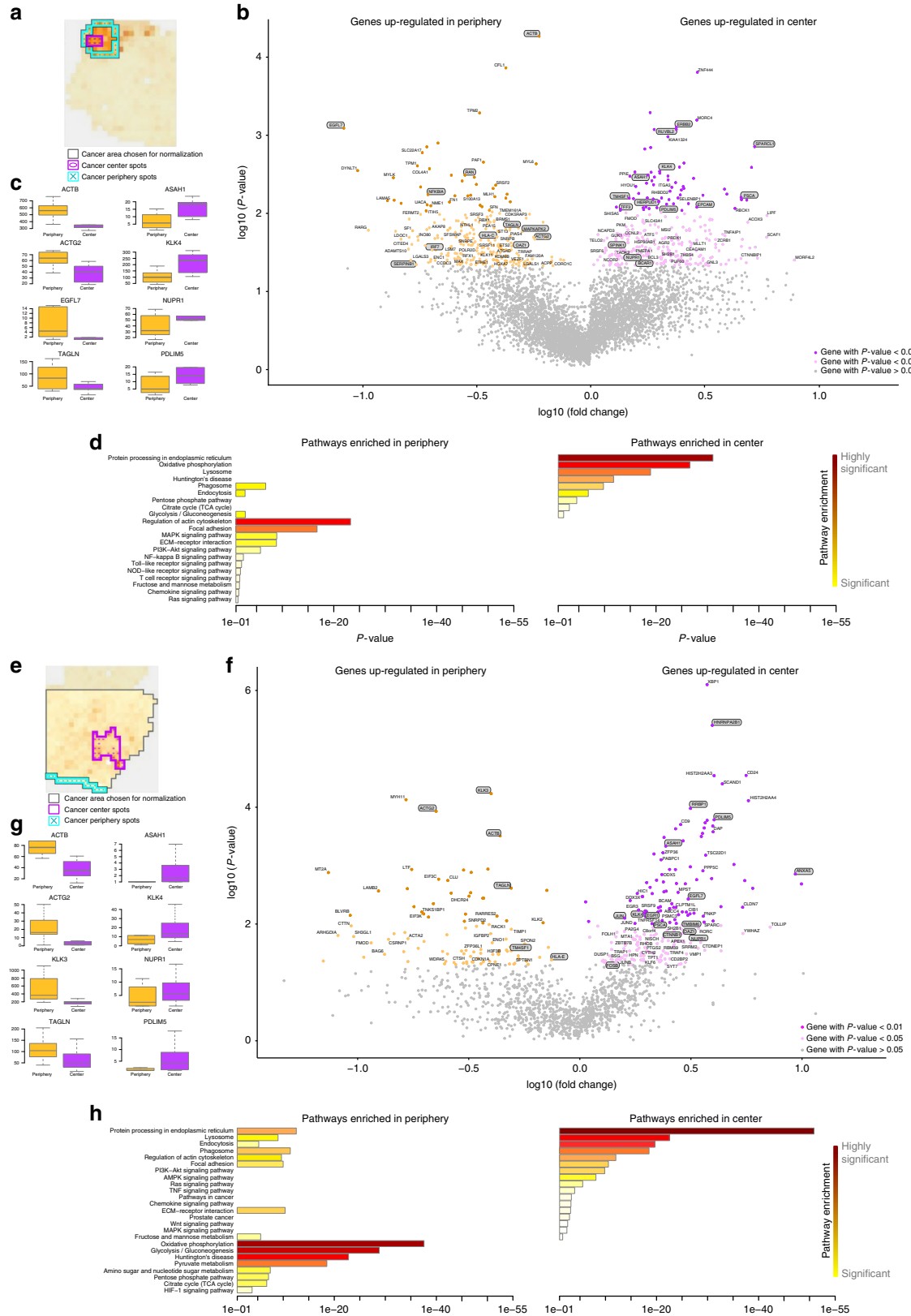

**Fig. 4** Spatial comparison of periphery and center of tumor transcriptomes. **a**–**d** Tissue sample 1.2 **e**–**h** Tissue sample 3.3 **a**, **e** Area comprising spots taken for normalization of ST counts, within this area spots are chosen as periphery and center. Choice of spots is based on the pathologist's annotation and the activity of the factors "cancer" and "reactive stroma" **b**, **f** Volcano plot of significantly differentially expressed genes between periphery and center **c**, **g** Box plots showing expression levels of noteworthy genes significantly upregulated in either periphery or the cancer center **d**, **h** Enriched pathways for significantly ($p < 0.05$) differentially expressed genes in center and periphery. $P$-values per gene were calculated with a two sample $t$-test

are *EGR1* and *KLK3* in sample 3.3 (Gs 3 + 4). *EGR1* is up-regulated in the center of the cancer area of sample 3.3 and there is evidence suggesting that this gene is directly linked to the transition of the cancer into invasive carcinoma and a potential target for cancer treatment[47,48]. *KLK3* is to date the best prostate cancer biomarker. However, its downregulation as observed in sample 3.3 (Gs 3 + 4) is linked to more aggressive tumors and recent studies have shown that the stimulation of *KLK3* expression can indirectly reduce the proliferation rate and decrease the risk of metastasis[49].

The cancer area of sample 2.4 shows similarities with the one in sample 1.2 (Fig. 4 and Supplementary Fig. 18). The center of 2.4 is dominated by enriched pathways linked to altered cellular metabolism (Glycolysis/Gluconeogenesis, Pyruvate metabolism, Amino sugar and nucleotide sugar metabolism)[38], whereas the periphery is marked by enriched pathways involved in stress, inflammation and immune system (B cell receptor signaling, T cell receptor signaling, Tolllike receptor signaling)[35,36]. Consistent with the factor intensities, the cancer area of sample 2.4 shows more pathway similarities with sample 1.2 than with sample 3.3 (Fig. 4 and Supplementary Fig. 18). The pathway Protein processing in endoplasmic reticulum is the most significantly enriched pathway in all three cancer centers. Due to increased cellular metabolism and cell proliferation rate, cancer cells activate this pathway to obtain correct protein synthesis and modifications and overcome ER stress[50].

**Spatial expression patterns in the microenvironment of cancer and inflammation**. To examine whether we could identify inflammation, we performed analysis of all 12 samples. The data revealed a factor ("inflammation" factor, Supplementary Fig. 19, Supplementary Data 6) that was active across inflammation-annotated regions of sample 3.1 and 4.2. A more granular factor analysis of only these two samples decomposes several stromal and glandular factors (Fig. 5, Supplementary Fig. 20, Supplementary Data 7). We detect spatial proximity of areas consisting of apparent cancer glands ("FOSB-enriched"), inflammation ("AQP3-enriched") and reactive stroma ("NR4A1-enriched") (Fig. 5b). A novel gene (*NR4A1*) was highly expressed in reactive stroma. We confirmed the spatial expression pattern of *NR4A1* by its protein by IHC (Supplementary Fig. 21). Among the factors not shown in Fig. 5, some contained stroma cells or normal glands (with or without *MSMB*) (Supplementary Fig. 22). *KRT13* was upregulated in one of the factors and has recently been associated with poor prognosis in metastatic patients[51].

Interestingly, both samples exposes stromal expression gradients adjacent to tumor regions (Fig. 5c) which are likely due to stromal cell-type heterogeneity. For the inflammation containing samples, hierarchical clustering of the factors revealed three distinct groups (Supplementary Fig. 22b). Stroma and normal cells separate from inflammation, however, compared to the previous clustering (Supplementary Fig. 2b, Supplementary Fig. 6b) reactive stroma is more similar to inflammation and cancer glands ("FOSB enriched") than stroma cells.

Whereas molecular alterations drive the progression from low-Gleason grade to invasive cancer, tumor microenvironment and tumor cells are co-dependent and progress alongside. Hence, we compared both the gene expression on stroma close to tumor and inflammation with that of normal stroma (Fig. 5d, Supplementary Data 8-9). The results display different enriched pathways between the two regions. Normal stroma was associated with cell movement and adhesion (actin cytoskeleton and regulation of Actin-based Motility by Rho), as well as androgen signaling and the complement system. Several studies have suggested that the

complement system is involved in the immuno-surveillance against tumors (anti-tumor effect)[52]. On the other hand, the complement has also been implicated in tumor growth[53]. Our results elucidated that the complement pathway is enriched in stroma cells close to normal epithelium. The reactive stroma was enriched for oxidative stress and ILK signaling. Studies have shown that ILK expression and activity is significantly up-regulated in several types of cancers (pro-tumor effect)[54].

To further study cancer and inflammatory microenvironments, we selected samples based on activity of the "reactive stroma" factor in the 12 sample analysis (Supplementary Fig. 19), Supplementary Data 6). A factor analysis of the four identified samples, and subsequent dimensionality reduction, reveal once again cancer, reactive stroma and inflamed glands in close proximity to each other (Supplementary Figs. 23–25, Supplementary Data 10).

In order to enable other reserachers to study their gene of interest on the tissue, a Shiny application was built and is freely available at https://spatialtranscriptomics3d.shinyapps.io/STProstateResearch/ (Supplementary Methods).

**Deletions and amplification are spread locally**. In order to complement the spatial gene expression patterns, we examined the copy number from whole genome sequencing data (Supplementary Fig. 26). We analyzed the affected base pairs in exonic regions per copy number values below two for each sample to gain insights about the genetic structure of each sample and to link deletions to cancerous areas. Sample 3.3 (Gs 3 + 4) shows more deleted base pairs above a CNV of 0.8 (15.6 kbp affected). We conclude that up to 50% of the cells in the tissue are affected by deletions if homozygous deletions are assumed. For sample 2.4 (Gs 3 + 4 and suspected Gs), higher number of deleted base pairs occurs with copy number of 1.3 (18 kbp affected) which appropriates 25% of the cells if only homozygous deleted segments are considered. The number of cells with deletions in the samples 2.3 and 3.1 is 35% and 25% respectively. Sample 1.2 (Gs 3 + 3) shows no higher number of deleted base pairs. None of the samples considered as histological normal show an increased occurrence of deleted segments. The samples that show an increased occurrence of deleted base pairs in exonic regions per copy numbers contain either cancerous or larger inflamed areas.

A similarity tree based on Euclidean distance and hierarchical clustering was created (Supplementary Fig. 27). The tree reflects the relation of deleted and amplified segments within the whole genome of the twelve tissue sections. Four clusters were revealed and each cluster contains one cancerous sample. We conclude that the genetic structural variations appear unique for each cancer sample. Sample 3.3 shows the largest difference to germline and contains the highest number of genetic structural variations of the twelve samples. Interestingly, we observe that the four cancer samples do not share many deletions or amplifications pointing to independent tumor origins. We compared the samples and their assigned clusters with their physical position within the prostate (Supplementary Fig. 27). For three clusters we observe that the samples, which belong to one cluster, are physically close to each other. It can be concluded that the deletions and amplifications are mostly spread locally.

Finally, we sought to assess the relationship of copy number and gene expression. Concerning amplifications and deletions, each cancerous or inflamed tissue sample shows a unique genetic structure (Fig. 6). In general, sample 1.2 is rather shaped by expressed genes with amplifications, sample 3.3 by deletions, and sample 2.4 harbors both, deletions and amplifications. Further, genes with amplifications or deletions are expressed primarily in small regions within the tissue mirroring a genetically hetero-geneous tissue.

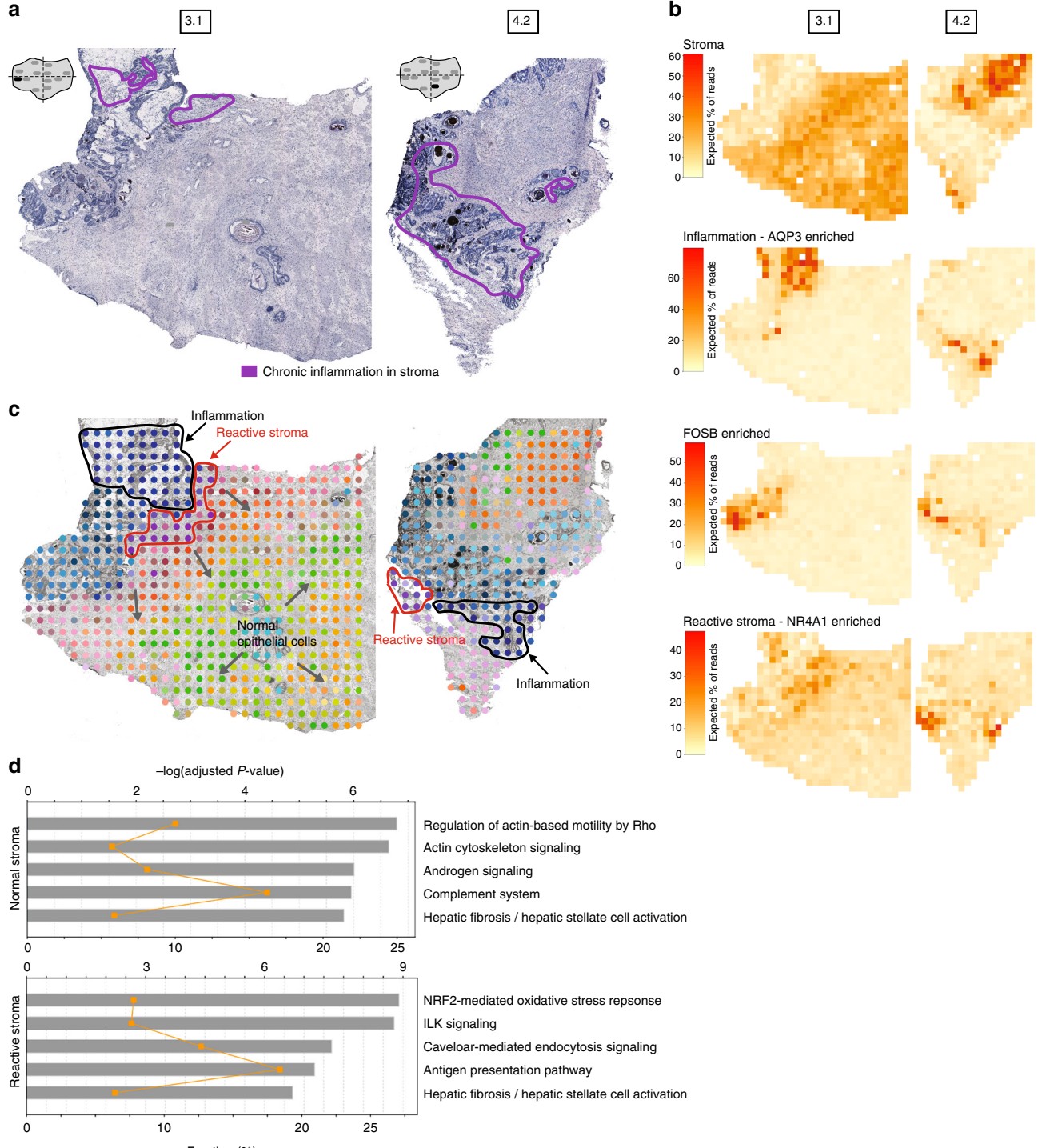

**Fig. 5** Stromal heterogeneity and reactive stroma in the microenvironment of inflammation. **a** Annotated brightfield images of H&E-stained tissue sections. **b** Selected factor activity maps of two inflammation-related factors and two stromal factors (normal and reactive) based on a joint factor analysis of tissue sections 3.1 and 4.2. Remaining factors' activity maps in Supplementary Fig. 22 and Supplementary Data 7. **c** t-SNE summary of factor activities of all factors from the analysis in **b**; similar colors indicate similar factor activities. Arrows indicate some stromal expression gradients. **d** Top five enriched pathways in reactive and normal stroma. Bars give significance and orange squares the ratio. P-values were corrected for multiple testing by the Benjamini-Hochberg procedure

## Discussion

Here we investigate tissue-wide gene expression heterogeneity throughout a multifocal PCa using the ST technology which quantify an array of transcriptomes across a tissue section. To ensure that patient genotype do not confound the analysis, we have selected a single prostate that is analyzed in a comprehensive manner, analyzing >6000 tissue regions. Compared to the spot diameter, the lateral RNA diffusion is negligible under the tissue slide, ensuring that the measured gene expression stems from the local tissue. The number of mRNA molecules obtained is in line with previous reports[23], and on average we sequence 2 million unique reads on a tissue slide. Hence, we measure, for the first

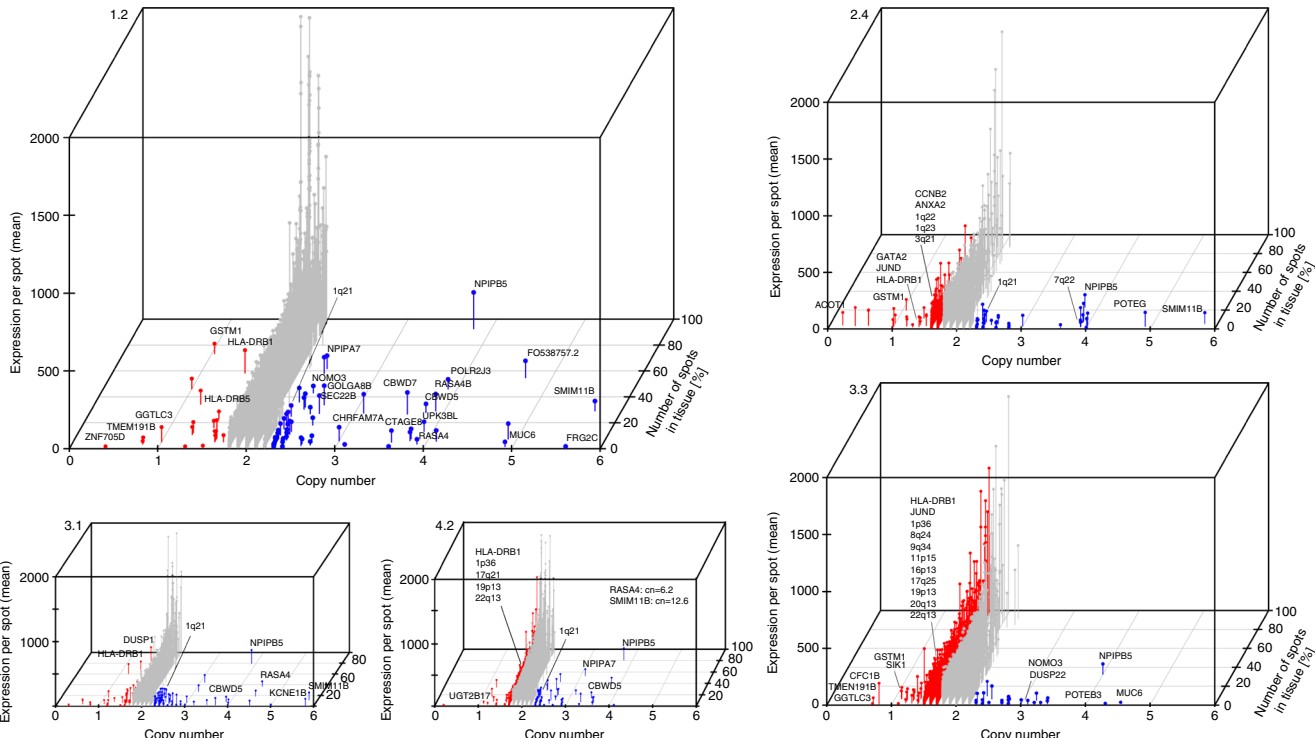

**Fig. 6** Relationship of copy number and gene expression. Expressed genes with a copy number between zero and six are shown. The majority of expressed genes have a copy number of two. Sample 1.2 is characterized rather by amplifications, sample 3.3 rather by deletions, whereas sample 2.4 displays a mixture of deletions and amplifications. Besides the deletions and amplifications that are unique for one or more samples, we identify several germline mutations. For example, we observe a heterozygous deletion of GSTM1 (CN = 1) which is linked to "an increased susceptibility to environmental toxins and carcinogens"[76] and a germline deletion at 11q11 (CN = 0) which is linked to obesity[77]. Somatic deletions were found among all 12 samples in USP17L8 (CN < 0.5)

time, spatial gene expression in PCa tissue sections. Further evidence for the accuracy of the spatial gene expression measurements is presented by concordance between gene expression and staining of the corresponding protein within the tissue.

We have developed an unsupervised probabilistic framework to analyze spatial transcriptomics data. The strength of this method lies in that the method takes the spatial position into account, which is lost in other used methods such as PCA and t-SNE. We identify factors corresponding to the different parts of tissue architecture, such as cancer, inflammation, normal or reactive stroma and normal or PIN glands, etc. Interestingly, we observe that distinct cancer expression regions can extend beyond the boundaries of annotated tumor areas. Similarly, the discovered gene expression profiles may be used to predict further regions of potential cancer, PIN or inflammation. From a clinical perspective, we suggest that our analysis may be used to alert pathologists to give extra attention to "high risk" areas based on localized, transcriptome-wide data. Furthermore, we observe clear separation of gene expression patterns between normal prostate epithelium and cancer areas with elevated Gs (3 + 3 and 3 + 4), as expected[55], suggesting that this approach may be useful to gain further understanding of human tumor in situ. Importantly, we report on heterogeneity within one patient as well as heterogeneity between patients. Thus, this study highlights the value of focusing on individual profiling.

The current investigation also provides new insights into gene expression differences between cancer core and periphery and pose significant questions that have important implications for the development of prostate cancer. A related question is whether the periphery of the tumor promotes tumor initiation and progression and if so may provide new tools for early detection.

Finally, we argue that a new landscape is revealed by spatially mapping gene expression and analyzing it in an unsupervised manner. For instance, it enables de-novo characterization and delineation of reactive stroma in the proximity of cancer and inflammation (Fig. 5) to elucidate the role of the microenvironment for PCa. Our study extends previous suggestions that inflammation and reactive stroma are evident at the earliest stages of neoplastic progression, stimulating development of the tumor[56] to demonstrate spatially confined gene expression gradients. Indeed, changes in the microenvironment may even precede genetic alteration in the tumor core. This study uncovers high levels of oxidative stress and ILK signaling within the reactive stroma whereas normal stroma was associated with cell movement and adhesion (actin cytoskeleton). High levels of oxidative stress in reactive stroma indicate that the cancer is dependent upon that the stroma releases energy to fuel cancer cells and enable growth and survival. Our spatial analysis describes reactive stroma as an emerging hallmark of cancer initiation and progression.

Taken together, our results have revealed that an analysis of tumor gene expression in a spatial context dramatically increases the granularity compared to a bulk analysis. Sampling different parts of the same tumor showed remarkable differences on the transcriptome level of the cancer cells at each site. Thus this massive tissue region analysis could serve as a foundation for a transcriptome-based clinical evaluation of cancer tissues as well as provide insight into gene expression in the tumor microenvironment. In summary, we propose that expression profiles based on spatial factor analysis can serve as a tissue-transcriptome dictionary as basis for a wide range of new scientific discoveries in prostate cancer.

## Methods

**Collection and preparation of prostate cancer tissue**. Radical prostatectomy was performed on a patient with adenocarcinoma (Gs 3 + 4, pT3b, PSA 7.1). The whole prostate was snap-frozen in liquid nitrogen and twelve sections covering half of the prostate were cryosectioned at 10 μm thickness. Sections were placed on prepared glass slides and incubated at 37 °C for 1 min, fixated in 16% methanol-free formaldehyde (#8906, Thermo Fisher Scientific) and washed in PBS. We applied the current ST protocol to all sections, yielding both traditional H&E images amenable to annotation by pathologists, as well as gene expression profiles for every microarray spot[23]. Tissue sections were annotated for pathological status and four were judged to exhibit tumor foci. Three of the tumor foci are situated in tissue regions covered by array spots.

The study was performed according to the Declaration of Helsinki and Good Clinical Practice. The study was approved by the Reginal Ethics Committee before study start (Application number, Dnr 2013/1869-31/1). All subjects were provided with full and adequate verbal and written information about the study before their participation. Written informed consent was obtained from all participating subjects before enrollment in the study.

**Preparation of quality control arrays and spatially barcoded arrays**. In short, for quality control tests poly-T20VN oligonucleotides (IDT) were consistently spread onto Codelink activated microscope glass slides, as per guidelines by the manufacture. Array production for experiments with spatial investigation was described previously[23]. The arrays were designed to have 1007 unique barcoded oligonucleotides with poly-T20VN capture areas. They were printed in areas of size $6200 \times 6600 \ \mu m^2$ on Codelink activated glass slides with a total of 1007 single spots. In order to keep the orientation, a frame with oligonucleotide (Eurofins) was printed as a border around the barcoded oligonucleotides, containing 148 single spots. Printed spots had a diameter of 100 μm and a center-to-center distance of 200 μm (center to center) from each other.

**Staining and imaging**. These steps were described previously[23]. Tissue was incubated in Mayer's Hematoxylin for 5 min and in eosin (1:20 in Tris buffer, pH 6) for 20 s.

**Quality control array experiments and detection of fluorescent cDNA "footprint"**. Prior to spatial barcoding experiments, permeabilization conditions were optimized in a quality control experiment for maximal mRNA yield in PCa tissue (Supplementary Fig. 28). The fluorescent cDNA was consistent with the tissue structure shown by histology.

Quality control experiments were carried out as described in section permeabilization and reverse transcription, except for some minor changes. This test was performed in order to study the optimal permeabilization conditions. The array was printed to have all sequences necessary for capture of mRNA but without a spatial barcode. The release step of the surface probes was not performed and the reverse transcription mixture contained the same reagents except for 0.5 mM of each dATP/dGTP/dTTP, 12.5 μM dCTP and 25 μM Cy3-dCTP. The procedure involved testing times of 8, 10, and 12 min. Glass slides were scanned in an InnoScan 910 scanner (Innopsys) with 5 μm resolution and gain at 10%. Signal intensities were investigated using the Mapix software.

**Permeabilization and reverse transcription**. These steps were described previously[23]. Exonuclease pre-permeabilization was performed at 37 °C for 30 min, followed by an incubation in pepsin at 37 °C for 10 min. Reverse transcription was performed at 42 °C overnight. In order to remove and degrade the prostate tissue, a mix of 1% β-mercaptoethanol in RLT buffer was added to the samples and incubated at 56 °C for 1 h and 15 min. A second removal mix containing Proteinase K in PDK Buffer was added to each well and incubated at 56 °C for 1 h and 15 min. The release step was done at 37 °C for 1 h and 15 min. Once the surface probes were de-attached, 65 μl from each well was collected. After probe release, the features with non-released DNA oligonucleotide fragments were detected by hybridization and imaging, as previously described[23], in order to obtain Cy3-images for alignment. The bright field images and fluorescent images were manually aligned using Adobe Photoshop CS6 (Adobe) by utilizing the visible spots and structures from both images.

**Library preparation of cDNA for sequencing**. The steps were performed as earlier described[23]. Finished libraries were diluted to 4 nM and sequenced on the Illumina HiSeq or NextSeq platform using paired-end sequencing. Typically, 31 or 101 bases were sequenced on read one to determine the spatial barcode, and 121 or 101 bases were sequenced on read two to cover the genetic region. ST sequencing reads were mapped against the human genome (GRCh38), and Ensembl (release 85) transcripts were quantified, as described previously[57].

**Dissecting spatial gene expression patterns with factor analysis**. We have developed a core model to perform factor analysis applicable for spatial gene expression data. The full model is described in Supplementary Note 1. This method (unsupervised) needs no prior knowledge of reference expression data. It seeks to

factor the gene expression into spatial factor activity maps and gene expression profiles. The factor activity maps reflect the amount of mRNA contributed by a given factor in every spot and are useful for visual inspection and comparison to morphological features. The expression profiles quantify how strongly each gene is expressed in a given factor and are thus informative about biological processes. The discovered "factors" correspond to cell types, but aside from biological effects, the analysis method also captures technical effects (such as sequencing depth of the libraries) and can allow for correction of such unavoidable artifacts. This enables us to quantify gene-expression differences between individuals, genotypes, developmental stages and disease states. Throughout the manuscript we name factors according to histological features co-incident with a factor's spatial activity or according to highly- and specifically-expressed genes.

**Poisson factorization core model**. Here, we give a high-level description of the core model to perform factor analysis on count matrices such as applicable for spatial gene expression data. The mathematical and computational aspects of the full model, including some extensions, are described in a separate supplement on mathematical methods (Supplementary Note 1).

We assume that the observed count $x_{gs}$ for gene $g$ in spot $s$ is the sum of count contributions $x_{gts}$ due to $T$ different factors ("cell types"), $x_{gs} = \sum_{t=1}^{T} x_{gts}$, and that these in turn are Poisson distributed, $x_{gts} \sim \mathrm{Pois}(\mu_{gts})$. The Poisson rate parameter $\mu_{gts}$ is the product of a gene- and type-dependent gene expression value $\phi_{gt}$ and a type- and spot-dependent spatial activity value $\theta_{ts}$, $\mu_{gts} = \phi_{gt}\theta_{ts}$. Notably, the gene expression $\phi_{gt}$ is independent of the spot, while the spatial activity $\theta_{ts}$ is independent of the gene. Then, the Poisson factorization core model is given by

$$x_{gs} = \sum_{t} x_{gts} \qquad (1)$$

$$x_{gts} \sim \mathrm{Pois}\left(\mu_{gts}\right) \qquad (2)$$

$$\mu_{gts} = \phi_{gt}\theta_{ts}, \qquad (3)$$

where the distributions of the non-negative random variables $\phi_{gt}$ and $\theta_{ts}$ still need specification. A graphical representation of the Poisson factorization core model is displayed in Supplementary Fig. 27, and parameters are learned by Monte-Carlo Markov chain (MCMC) sampling.

In Poisson factorization the expected observations are given by the matrix product of the gene expression and spatial activity matrices, $\mathbb{E}[X] = \Phi\Theta$, because $x_{gs} = \sum_t x_{gts} \sim \mathrm{Pois}(\sum_t \mu_{gts}) = \mathrm{Pois}\left(\sum_t \phi_{gt}\theta_{ts}\right)$ and thus $\mathbb{E}[x_{gs}] = \sum_t \phi_{gt}\theta_{ts}$.

The full model is described in a separate mathematical methods supplement (Supplementary Note 1) and comprises extensions not mentioned here. The extensions include spot-dependent scaling variables, spatial smoothing, as well as capabilities to perform joint analyses for multiple samples.

**Summarizing and visualizing patterns of gene expression and factor activities**. In order to visually summarize spatial patterns present in thousands of genes or across multiple factors, we make use of t-distributed stochastic neighbor embedding (t-SNE)[58]. Specifically, we reduce matrices that have rows for every spot and columns for every gene (or columns for the activities of multiple factors) to matrices that have three columns so that these three dimensions can be used as coordinates in color space. When spots are colored in this way, similar colors indicate similar gene expression. Such colored spots are then overlaid on the histological image for a joint presentation of histologic and transcriptome-wide information.

**Hierarchical clustering for sample 1.2**. The bright field image was converted into grayscale in Photoshop and loaded in R using the "jpeg"-library. The image was converted to binary dots using the "base"-library. A virtual grid was generated across the tissue and each binary dot was assigned to a grid-cell using the "sp"- and "raster"-library. The values from the spatial spots were used to generate a linear interpolation, using the "Akima"-library, across the array. Hierarchical clustering was carried out based on Euclidean distance between the spots and each cluster was assigned a color. The colors were further assigned to all grid-cells and each binary dot was assigned the color that corresponded to the grid-cell it was localized in. Areas in between clusters were made transparent using the "scales"-library.

**Method description for factor trees**. The expression profiles of each factor were used to calculate the Jaccard distance between them[59]. Hierarchical clustering agglomeration method ward.D2 was applied to build the tree (R packages "vegdist", "ape", and "stats")[60–62].

**Gene expression analysis for sample 1.2**. Intra-replicates were extracted from each region (normal = green color, cancer = red color and PIN = blue color in Supplementary Fig. 4b) within the 1.2 cancer tissue sample, and contained between 4–5 spots. At least 3 sets of intra-replicates were created for each area of interest. Count data was generated with HTSeq-count (version 0.6.1)[63]. The -m parameter

was set to union and the count data was imported into the statistical software R. A heatmap was made of the most variable genes, across all regions' replicates, using regularized log (rlog) transformed count values (DESeq2, version 1.12.3)[64]. Principal components were calculated based on the 500 most variable gene counts after rlog transformation. Individual genes signatures were plotted in concurrence with their location on the spatial array. The sizes of dots are proportional to normalized counts per million (CPM) values for the specific gene for visualization purposes.

**Gene expression analysis for sample 3.3**. Gene counts were extracted from each spot belonging to the specified conditions (normal, cancer or PIN contained between 3–6 spots per replicate) of interest as determined by the ST analysis. The gene counts were imported into R and CPM values were computed. A matrix was created with rows corresponding to genes and columns corresponding to samples. Row variances were computed across all genes to extract the top 500 genes with highest variance across all regions' replicates. The CPM values for these genes were visualized in a heatmap (Supplementary Fig. 7, Supplementary Data 11).

**Immunohistochemistry**. Frozen tissue sections stored in −80 °C were thawed in RT to be fixated with 3% freshly made paraformaldehyde in TBS for 10 min in RT. Tissues were then permeabilized for 10 min in TBS+ 0.1%Triton-X100, rinsed three times in TBS for 5 min/ rinse. Blocking with 2% bovine serum albumin in TBS for 2 h was performed before the tissues were incubated with primary antibodies overnight at 4 °C. After rinsing with 3 × 5 min with TBS the tissues were incubated with the secondary antibodies donkey anti-mouse immunoglobulin G (IgG)-AlexaFluor 568 (1:500 Molecular Probes) and donkey anti-rabbit IgG-AlexaFluor 647 (1:500 Molecular Probes) for 1 h at RT in darkness. DNA was counterstained with DAPI (Molecular Probes) and slides were mounted with Prolong Gold (Molecular probes). Tissues were stained with antibodies against SPINK1 (1:50, H00006690-M01, 4D4, Novus), TFF3(1:200, HPA 035464, Sigma), SPON2 (1:100, A-10, st cruz), PGC (1:50, NBP1-91011, Novus), NPY (1:100, ab48789, Abcam), Aquaporin (1:100, ab168387, Abcam), NR4A1 (1:100, ab48789, Abcam), ACPP (1:100, Biologicals), P63(1:150, ab53039, Abcam), Vimentin (1:150, ab8069, Abcam). Fluorescence images were obtained with a Zeiss LSM 780 inverted confocal microscope, using a Plan Apochromat 20 × /NA (numerical aperture) 0.7 objective. Tiled images were acquired from optical sections of 5 micrometer.

**Pathway analysis in Fig. 3**. The gene expression profiles of each factor are the basis for the pathway annotation. We performed outlier detection genewise of the normalized expected values to extract differences in expression between the factors. For each gene in each factor a $z$-score was calculated based on fitting a normal distribution to the gene's expression in the other factors. We defined significant outliers as genes with a $z$-score > 2.5 and where the distribution passed the Komolgorov–Smirnov normality test at the 0.05 alpha level. The resulting gene list per factor was submitted to PathwAX[65] on the KEGG database[66] (Fig. 3c, Supplementary Data 4)

**Comparison of expression in cancer center and cancer periphery for samples 1.2, 2.4, and 3.3**. We used the normalized ST-counts per gene and spot. Starting from the size of the area where the "cancer" factor is active, we chose 42 spots (sample 1.2), 403 spots (sample3.3), and 17 spots (sample 2.4) respectively, comprised of this area.

To ensure quality we removed spots with a log-library size lower than 3 median absolute deviations below the median log-library size (R package "scater")[67]. Additionally, we removed low-abundance genes with zero or near-zero counts. The filtered data set is normalized using the deconvolution method which is based on pool-based size factors and the assumption that most genes are not differentially expressed. The counts from cells were pooled to calculate the size-factors which are used for a cell-specific normalization (R package "scran")[68]. We utilize the quickCluster method from the R package "scan" to identify optimal pool sizes.

The resulting normalized counts per gene and spot within the tumor region were used to compare expression in the tumor periphery with the center.

Spots located in the periphery and in the center were defined based on the pathologist's annotation and the active cancer factor. Spots with mainly stroma cells were removed. The fold change per gene was calculated as gene expression mean of center spots divided by gene expression mean of periphery spots. $P$-values per gene were calculated with a two sample t-test[69] at confidence level 0.95. Genes with a $p$-value < 0.05 were submitted to PathwAX[65] on the KEGG database[66] (Supplementary Data 12–14).

**Pathway analysis in Fig. 5**. Ingenuity Pathway Analysis software (Build version 456367 M, Content version 39480507 release date 20170914) (Ingenuity Systems, Redwood City, CA) was used to identify significantly enriched pathways. To calculate significance of enrichment (Fisher´s exact test, performed within the software) the reference molecule set was Ingenuity Knowledge Base (Genes only). Input data was extracted from the "Stroma—PTGDS enriched" and "Reactive stroma" factors in Fig. 5, using the top 200 genes (Supplementary Data 8 and 9).

**Shiny application**. The HE and Cy3 images were aligned and the spots under the tissue stain detected. These spots' coordinates and the corresponding transcript counts were used in the further analysis and image processing. In order to ensure quick data visualization, a Shiny application was built and is freely available at https://spatialtranscriptomics3d.shinyapps.io/STProstateResearch/. The application visualizes spatial gene expression as an interpolation of a regular grid[70] of the coordinate points previously discussed. Then, a tissue mask is placed on top of the interpolated grid to create the final heatmap image presented in the application.

Shiny application access information:

Login with a Google account: Username: 3dstresearch@gmail.com, Password: AIK2017!

**Extraction and fragmentation of DNA**. DNA was extracted from adjacent sections to each of the twelve sections used for the spatial barcoded array. In order to give a total amount of 100 µm, a total of five sections per bulk sample were cryosectioned at 20 µm. PCa tissue was put in Lysing Matrix D tubes (#116913050, MP Biomedicals) and homogenized in a FastPrep (MP Biomedicals). All the samples were then prepared with the AllPrep DNA/RNA Micro Kit (#80284, Qiagen). DNA extracted from blood was included as germline control using the Gentra Puregene Blood kit (#158445, Qiagen).

**Library preparation for whole genome sequencing**. Whole genomes libraries were made from extracted DNA (both from tissue and blood) using the NeoPrep Library Prep System (Illumina TruSeq Nano) according to the manufacturer's protocol. Libraries were sequenced with at least 30 × (tissue) or 42 × (blood) coverage on HiSeqX (HiSeq Control Software 3.3.39/RTA 2.7.1) with a 2 × 151 setup using HiSeq X SBS chemistry.

**DNA sequence alignment**. The reads of each sample were aligned with Burrows–Wheeler Aligner (BWA) against the human assembly GRCh38 (ensemble) release 84. BWA *mem* was performed since it is recommended for high-quality queries and longer sequences[71]. The reads in our whole genome sequence (WGS) data have a read length of 151 bp. Furthermore, we used Samtools[72] for converting sam to bam format, sorting, indexing, and for converting the alignments to bed files.

**Copy number calling**. Copy numbers were inferred with the R package "Read-Depth"[73] based on WGS data of the twelve tissue samples, which was aligned with BWA. Firstly, "ReadDepth" was applied with default values but with annotations computed for our read length of 151 bp and GRCh38. The copy number values were inferred for each sample independently. Secondly, we matched the segments and corresponding copy numbers inferred by "ReadDepth" to GRCh38 release 84 to compare copy number variations (CNVs) in coding regions. We filtered all exons of protein coding genes from the reference genome. If start and end position of an exon is located within a segment, we applied the corresponding copy number. In the rare case of two or more segments that span an exon (1608 out of 343,705 exons; 0,46%), we assigned two (1606 exons) or more (2 exons) copy numbers, however, considered the segment length per copy number within these exons.

For the CNV analysis we needed to know which copy number value is normal. In our samples the peak of base pairs per rounded copy number value for each sample is at 1.8. Therefore, we set a CNV of 1.8 as normal instead of 2.0. The chromosomes X and Y were excluded since here the normal copy number value is one and the borderline for deletions and amplifications is different from the autosomes.

**Similarity tree building**. The smallest bin size of the calculated segments and copy numbers by "ReadDepth" is 1200 bp for each of the twelve samples. In other words, each segment start, end, and length of the twelve samples is a multiple of 1200 bp. Therefore, we sliced the genome in segments of 1200 bp length and assigned the corresponding copy number values of each sample. This results in ~2.5 Mio segments (3088 Mbp genome length/1200 bp) and a copy number value vector per segment containing one copy number for each sample. To build the tree we filtered for clear deletions and amplifications by only accepting the CNV vector as input data if the CNV was below 1.6 or above 2.3. Furthermore, we excluded the 1200 bp segments that contain the centromere since centromeres are difficult to sequence reliably. We also excluded the X and Y chromosomes since their normal CNV value is one. Finally, only unique CNV vectors were chosen to reduce the data space. The data set for the tree contains 287 unique CNV vectors. The tree was built with the R package "Pvclust"[74] based on Euclidean distances, the hierarchical clustering agglomeration method ward.D2, and 1000 bootstraps.

**Copy number variation correlation with gene expression**. Copy numbers were inferred as described earlier with he R package "Readdepth" and the segments were mapped to GRCh38 release 84. To display clear deletions and amplification, the copy numbers were corrected by a scaling summand of 0.2 in order to recenter the values (compare Copy number calling). Genes with a break point were excluded.

In addition, the chromosomes X and Y were excluded since their normal copy number is one. The expression spot mean is presented as relative to the sample

expression spot mean and cut at 2000% to emphasize genes with a copy number below or above two. The figures were generated with R package "Scatterplot3d"[75].

**Probes and primers**. Surface reverse transcription oligonucleotide for quality control experiments

[AmC6]UUUUUGACTCGTAATACGACTCACTATAGGGACACGACGCT CTTCCGATCTNNNNNNNNTTTTTTTTTTTTTTTTTTTTVN

Surface reverse transcription oligonucleotides with spatial barcodes:

[AmC6]UUUUUGACTCGTAATACGACTCACTATAGGGACACGACGCTC TTCCGATCT[18mer_Spatial_Barcode_1to1007]WSNNWSN

Surface frame oligonucleotide:

[AmC6]AAATTTCGTCTGCTATCGCGCTTCTGTACC

aRNA ligation adapter:

[rApp]AGATCGGAAGAGCACACGTCTGAACTCCAGTCAC[ddC]

Second reverse transcription primer:

GTGACTGGAGTTCAGACGTGTGCTCTTCCGA

PCR primer InPE1.0 primer:

AATGATACGGCGACCACCGAGATCTACACTCTTTCCCTACACGACG CTCTTCCGATCT

PCR primer InPE2.0 primer:

GTGACTGGAGTTCAGACGTGTGCTCTTCCGATCT

PCR Index primer:

CAAGCAGAAGACGGCATACGAGATXXXXXXGTGACTGGAGTTC

Cy3 anti-A probe:

[Cy3]AGATCGGAAGAGCGTCGTGT

Cy3 anti-frame probe:

[Cy3]GGTACAGAAGCGCGATAGCAG

**Code availability**. The factor analysis software is available under the GNU General Public License v3 at https://github.com/maaskola/spatial-transcriptome-deconvolution.

**Data availability**. Count matrixes are available at http://www.spatialtranscriptomicsresearch.org/. Sequencing data are deposited at the European Genome–Phenome Archive (EGA), hosted by the European Bioinformatics Institute (EBI), under the accession number EGAS0000100300

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

## Acknowledgements

This study was supported by AstraZeneca and Science for Life Laboratory. We thank National Genomics Infrastructure (NGI), Sweden for providing infrastructure support. The data were analyzed using resources provided by SNIC through the Uppsala Multidisciplinary Center for Advanced Computational Science (SNIC/UPPMAX). We thank Annelie Mollbrink for providing lab assistance, Mobashir Mohammad and Sailendra Pradhananga for performing inital analyses, and Johan Lindberg, Pelin Sahlén and Carsten Daub for comments on the manuscript.

## Author contributions

Experiments were conceived and designed by E.B. Experiments were performed by E.B., N.S., K.W. and M.M. Computational methods were developed by J.M., with input from J. Lagergren. Bioinformatics analyses were performed by E.B., J.M., S.F., C.O., L.L., F.S., S. V., and J.B. A.T. inspected the tumor samples and performed morphological annotations. F.T. provided samples and biological input. The manuscript was written by E.B., J.M., T. H., S.F., E.S., and J.L. with input from all authors.

## Additional information

**Competing interests:** P.L.S., F.S., and J.L. are authors on patents applied for by Spatial Transcriptomics AB covering the technology. The remaining authors declare no competing interests.

