## [Peer Review File · Nature Communications]

Reviewers' comments:

Reviewer #1 (Remarks to the Author):

Intra-tumor heterogeneity has so far been studied either by comprehensive analysis of cellular diversity without spatial information, or by spatial analysis with only selected genes. Thus, the ability to combine the comprehensiveness of transcriptome profiling with spatial information represents a highly important challenge in the field, with attempts by multiple platforms and labs. One of the promising directions is the Spatial Transcriptomics developed recently and demonstrated in breast cancer. Here, Berglund et al. apply the method to prostate cancer. The approach is applied comprehensively to a single prostate tumor, combining pathology with spatial transcriptomics to show an overall agreement and extensive validations. Distinct tumor components are captured as "factors" in the analysis, such as cancer, stroma, and inflammatory regions. In further analysis, several spatial patterns are demonstrated, including the discrimination between two distinct stromal and inflammatory areas.

This work demonstrates the ability to apply Spatial transcriptomics to prostate cancer and to improve the depth through which spatial tumor analysis may be performed. However, the manuscript is technical and difficult to follow, the results are limited to validations and a broad level of description of each component, and ultimately there appears to be limited insight derived from this work. As it stands the work is focused on validations of the approach, but does not demonstrate a significant advance in understanding spatial patterns in cancer and might be appropriate for a more specialized journal.

Major Comments:

1. In all of the analysis, there is no use of the theoretical richness of the data, and the results only report few selected genes and generic classifications, lacking a more nuanced description of spatial patterns and expression programs. A transcriptome-wide measurement should enable the identification of various spatial patterns reflecting diversity of the same cell type or of the interactions between cell types. It would seem then that further insight may be derived from the data, unless quality of the data is poor. It is unclear to me if the limited results in that regard reflect limitations in the data or in the analysis, but I would expect a more detailed descriptions of transcriptome diversity in order for this work to be considered as a significant contribution.

2. I found the manuscript difficult to follow, and overly technical, thereby limiting the ability to appreciate the results. I would suggest the authors to both extend their analysis (as noted above) and simplify the descriptions of selected analysis, focusing on concrete results beyond those which largely validate the method but add limited insight.

3. The manuscript reflects a very comprehensive spatial analysis, and therefore it would be sufficient to have only few tumors analyzed. Yet, a single tumor would still seem insufficient. Adding few other tumors may enable further results to be derived by integrating the observations across tumors.

Reviewer #2 (Remarks to the Author):

This manuscript describes the utilization of a novel spatial sequencing technology developed by the same group on measuring global gene expression in prostate cancer. Based on their landmark paper (Science, 2016), here the authors develop a strategy to characterize tissue regions on the basis of the cellular level without distorting spatial information. They develop a new (factor analysis-based) algorithm for the delineation of tumor and stromal tissue based on the regional gene expression patterns. As a result, they are able to describe the tumor heterogeneity with so far unprecedented resolution. Furthermore, the authors compare their data with histological information of adjacent tumor slices from the same patient.

The study is performed well and the data are presented clearly. The amount of data and the resolution of the spatial gene expression are impressive. It is not fully unexpected in the post-genomic era that huge tumor heterogeneity was found. My major criticisms are that samples from only one patient were analyzed and that the results presented remain descriptive and correlative. The gain of knowledge about tumor biology and its dynamics from this kind of data is limited. However, this reviewer feels that the power of the technology as well as the novel way of data analysis presented prevail the critical points. Once published, it can be expected that the manuscript will be highly cited by scientists working in the fields of the prostate cancer and tumor heterogeneity. Therefore, I recommend that this manuscript be accepted for publication in Nature Communications.

N.B. Although this reviewer understands the principle of the factor analysis used, an expert statistician would be more suited to evaluate the feasibility of this approach.

Replies to the reviewers' comments

All changes are highlighted in purple color in the manuscript.

Reviewer #1

Comment (i)

In all of the analysis, there is no use of the theoretical richness of the data, and the results only report few selected genes and generic classifications, lacking a more nuanced description of spatial patterns and expression programs. A transcriptome-wide measurement should enable the identification of various spatial patterns reflecting diversity of the same cell type or of the interactions between cell types. It would seem then that further insight may be derived from the data, unless quality of the data is poor. It is unclear to me if the limited results in that regard reflect limitations in the data or in the analysis, but I would expect a more detailed descriptions of transcriptome diversity in order for this work to be considered as a significant contribution.

Reply to Comment (i)

We fully acknowledge the comments from the reviewer. We have accordingly modified the paper to focus on some more substantial spatial findings.

In particular we have added a more thorough and detailed description of the transcriptional landscape by demonstrating differences between the cancer, PIN, and normal epithelium factors in Fig. 3. We performed pathway analysis (Fig. 3c) and found higher expression of cancer-related pathways in cancer and PIN compared with normal epithelium. We detected pathways associated with e.g. cell cycle, DNA replication, homologous recombination, and p53 signaling in the cancer and PIN areas. These represent the main biological differences between normal and tumor of prostate cancer at a molecular level.

Furthermore, we investigated differences between normal (close to normal epithelium) and reactive stroma (surrounding inflammatory tissue) by pathway analysis (Fig. 5d). Normal stroma was enriched for actin cytoskeleton (cell movement and adhesion) as expected. A reactive stroma gene signature reflected pathways associated with oxidative stress and ILK signaling (pro-tumor effect). These data confirm that the microenvironment plays an important role in tumor initiation and progression and that extensive intra-heterogeneity also exists within the stroma.

Importantly, we have also developed a shiny-App to allow any researcher to perform datamining for their genes of interest in the spatial context using our transcriptome wide dataset.

Comment (ii)

I found the manuscript difficult to follow, and overly technical, thereby limiting the ability to appreciate the results. I would suggest the authors to both extend their analysis (as noted above) and simplify the descriptions of selected analysis, focusing on concrete results beyond those which largely validate the method but add limited insight.

Reply to Comment (ii)

We have, in line with reviewer, extended the analysis (see point above) and tried to clarify the description of the analysis by giving a more simplified description of the data.

Comment (iii)

The manuscript reflects a very comprehensive spatial analysis, and therefore it would be sufficient to have only few tumors analyzed. Yet, a single tumor would still seem insufficient. Adding few other tumors may enable further results to be derived by integrating the observations across tumors.

Reply to Comment (iii)

We agree that more samples will strengthen the observations of the analysis across tumors and we have now performed spatial analysis on two more patients to investigate the tumor heterogeneity in prostate cancer (Supplementary Figs. 13-14). We observe high variability between patients. Still, we could integrate previous observations in one of the new tumors. Patient 3 also contained high levels of SPON2, NPY and NR4A1. Interestingly, patient 2 contained unique sets of expressed genes not observed in the other ones (Supplementary Fig. 14).

Reviewer #2

We would like to thank the reviewer for positive comments regarding our approach to analyze spatial prostate cancer transcriptomics data. We have in line with the reviewer's suggestion included two more patients to expand and support the observed diversity of the transcriptomic landscape in prostate cancer samples. We have now added the result of our experiment in Supplementary Figs. 13-14, and a section in the "Results" at the paragraph "*Spatial expression patterns common to cancer tissue sections*".

Reviewers' comments:

Reviewer #1 (Remarks to the Author):

The revised version is significantly improved with respect to two of my previous concerns. First, more samples have been added, and the current dataset is now clearly sufficient for publication. Second, the paper structure and division between main and supplement has been improved and it is now a much easier and interesting read.

However, my third concern relating to the computational analysis remains largely unchanged. I still feel that the paper lacks in the scope and results of the computational analysis. The focus is on a factor analysis and on comparison between elements that are clearly distinct (e.g. cancer vs. normal), and there is limited attempt to perform more subtle analysis and understand spatial differences within a given element (i.e. same cell type). I think it is important to avoid the feeling that results from spatial transcriptomics ultimately converge to (1) trivial differences (cancer vs. normal) and (2) an overall classification of region into well-defined types that could also be distinguished by much simpler methods such as staining with selected markers. The richness of the data should allow more subtle analysis and unexpected results, and these are still lacking. The authors should be best suited to identify what analyses might be relevant and feasible but I will try to highlight two suggestions.

1. The factor analysis defines 10 factors, but only a subset of these are directly discussed; what do the others represent? although this might not be trivial I believe it is worth more attention and a simple list of genes and enrichments is not sufficient. are there any cell types identified in the growing compendiums of single cell datasets that fit these un-annotated factors? could they be a combination of multiple interacting cell types? do they correlate with any histological parameter? if spatial transcriptomics ultimately leads to dataset that is best summarized by 10 elements then the authors should have a good understanding of each of those elements. if the results are inconclusive then these analysis could be in supp. but a more comprehensive attempt seems justified. similarly, if the parameters of the factor analysis are altered what would be the next additional factors?

2. even if the data is best summarized by 10 distinct factors, there are likely additional effects that relate to interactions between those factors. for example, in what way are cancer regions different when they are in the vicinity of stroma or when they are at the core of a dense cancer-rich area? to address such questions more generally, I would suggest the following analysis: First, identify all regions that correspond to a given factor/cell-type. then, examine the relationship with a second factor (one at a time): examine the expression of each gene in regions of factor X as a function of the region's distance from elements of factor Y, and look for non-random patterns, such as genes that increase as a function of adjacency to Y. genes that are themselves expressed by Y would obviously show up but can be excluded; even better, their pattern can be compared to the expected pattern from a weighted linear combination of X+Y factors (such that deviations might be detected of genes that go up in X more than would be expected by the pure contamination of cells from Y). in this way, the authors can begin to systematically address the influence of adjacent regions on one another. It might be that my suggestion is naive and reflects my limited understanding of this data, but I strongly feel that additional attempts should be made to extract more results from this data.

Reviewer #2 (Remarks to the Author):

The authors have addressed my previous points of concern.
I recommend the paper for publication in Nature Communications.

Replies to the reviewers' comments

All changes are highlighted in purple color in the manuscript.

Reviewer #1

Comment (i)

The factor analysis defines 10 factors, but only a subset of these are directly discussed; what do the others represent? although this might not be trivial I believe it is worth more attention and a simple list of genes and enrichments is not sufficient. are there any cell types identified in the growing compendiums of single cell datasets that fit these un-annotated factors? could they be a combination of multiple interacting cell types? do they correlate with any histological parameter? if spatial transcriptomics ultimately leads to dataset that is best summarized by 10 elements then the authors should have a good understanding of each of those elements. if the results are inconclusive then these analysis could be in supp. but a more comprehensive attempt seems justified. similarly, if the parameters of the factor analysis are altered what would be the next additional factors?

Reply to Comment (i)

We thank the reviewer for adding this perspective. We have added a more detailed description of all factors that were not discussed previously (Supplementary Figs. 3, 8, 21). Indeed, many of the factors are a combination of different cell types (normal, prostatic atrophy (mimic of cancer) or stroma) indicated by the hierarchical clustering of factors, morphological investigation, implicated genes, and calculated % of stroma and epithelial cells.

Interestingly, one factor contained high levels of MSMB, which is known to be highly expressed in normal prostate (particularly in high-grade tumours). Another noteworthy observation was that one factor seemed to represent cells that were not malignant nor normal (Supplementary Fig. 8). This factor had elevated levels of RACK1, a gene that is known to interact with the androgen receptor.

Since the weight of each factor decreases with the addition of new factors there is a balance to be considered between representing the full complexity of defined signatures and false positive characteristics or clustering of cell types. The number of clusters have thus been tailored to what appears to make the most sense given the expression data and pathological annotations.

We agree that single cell data from dissociated cells/nuclei from solid tumors could be useful for additional annotation. But we have not identified any single cell dataset that could explain the un-annotated factors and only a few datasets, of less relevance, on circulating prostate tumor cells have been published so far^{1,2}.

Comment (ii)

Even if the data is best summarized by 10 distinct factors, there are likely additional effects that relate to interactions between those factors. for example, in what way are cancer regions different when they are in the vicinity of stroma or when they are at the core of a dense cancer-rich area? to address such questions more generally, I would suggest the following analysis:

First, identify all regions that correspond to a given factor/cell-type. then, examine the relationship with a second factor (one at a time): examine the expression of each gene in regions of factor X as a function of the region's distance from elements of factor Y, and look for non-random patterns, such as

genes that increase as a function of adjacency to Y. genes that are themselves expressed by Y would obviously show up but can be excluded; even better, their pattern can be compared to the expected pattern from a weighted linear combination of X+Y factors (such that deviations might be detected of genes that go up in X more than would be expected by the pure contamination of cells from Y). in this way, the authors can begin to systematically address the influence of adjacent regions on one another. It might be that my suggestion is naive and reflects my limited understanding of this data, but I strongly feel that additional attempts should be made to extract more results from this data.

Reply to Comment (ii)

We thank the reviewer for adding this additional point. We investigated the interaction between the factors to achieve a more comprehensive understanding (Supplementary Fig. 3b, 8b and 21b) in line with the comments from the reviewer.

Hierarchical clustering of factors from analysis in Fig 2 (Supplementary Fig. 3b) revealed that stroma and reactive stroma grouped together. Notably, inflammation clustered close to PIN and cancer, suggesting that inflammation is an important contributor in cancer progression. Normal glands (with and without MSMB) clustered by their own as expected. Hierarchical clustering of factors from analysis in Fig. 3 (Supplementary Fig. 8b) appeared very similar as the one in Supplementary Fig. 3b.

We also investigated differences in gene expression in the periphery of the cancer and neighbouring regions. We found three main clusters and identified genes that were highly expressed both within the cancer as well as close to the cancer, but decreased with increased distance from the cancer edge (Supplementary Figs. 6a, b, c).

A more fine-grained analysis as suggested is not possible for those samples containing several activated factors in direct vicinity of tumor. We therefore performed for a simpler approach by comparing the expression in groups of spots at the center and at the periphery of a cancer-rich factor to analyze which genes and pathways contribute most to the progression of the disease (Supplementary Figs. 6d and e). Briefly, in the periphery we identify genes linked to inflammation, cell growth and differentiation, and cell motility while in the center we see more genes related to metabolism and glucose consumption. This suggests, as expected, that the center contains cells with high energy consumption due to rapidly growing cells. Pathway analysis of differentially expressed genes between these areas emphasizes this conclusion. We have extended the manuscript with these results.

References

1. Miyamoto D. T., Zheng Y., Wittner B. S., Lee R. J., Zhu H., Broderick K. T. RNA-Seq of single prostate CTCs implicates noncanonical Wnt signaling in antiandrogen resistance. *Science*. 349, 1351–1356 (2015).
2. Hwang, J. E., Joung, J. Y., Shin, S. P., Choi, M. K., Kim, J. E., Kim Y. H., Park W. S., Lee S. J., Lee K. H. Ad5/35E1aPSESE4: A novel approach to marking circulating prostate tumor cells with a replication competent adenovirus controlled by PSA/PSMA transcription regulatory elements. *Cancer Lett.* 372, 57–64 (2016).

Reviewers' comments:

Reviewer #1 (Remarks to the Author):

The revised version is improved and I believe that the core/periphery analysis adds to the paper. However, I also have several remaining comments regarding this new analysis as it currently does not seem to be solid enough and it would seem like more thought should be given to the analysis, interpretation and presentation of these results.

1. The initial clustering shown by Fig. 6A is not reliable, as the tSNE plot does not indicate three distinct clusters. Of course, clustering algorithms will always produce some definition of clusters, but I would expect that a plot that demonstrates the clustering should show a pattern that supports the clustering, which is not the case in this figure. If the clustering is indeed robust, then perhaps the choice of presentation is suboptimal and it could be shown in another way (e.g. with a dendrogram). Otherwise, if the clustering is forced by the use of a clustering algorithm and the signal for this clustering is not robust then I would suggest using a different computational approach, such as PCA or NMF, which may deal better with a continuous rather than discrete pattern, or to completely eliminate the clustering analysis.

2. Are the results from the core/periphery analysis consistent between tumors? This analysis appears to include only one tumor, and while I understand that there is much more data for this compared to the other tumors, I do believe that it is important to make sure that there is some consistency with at least one other tumor, and perhaps report only the consistent effects. If the data for the other tumors is insufficient to perform a similar analysis, then perhaps the authors can just evaluate the effect - as defined in the first tumor - in the context of other tumors. However, a separate analysis in each tumor, followed by integration of the results across tumors, would be better.

3. Some of the signal in the periphery involves immune-related genes. It is difficult to conclude if these are due to contamination of these regions by few immune cells or if this is truly reflecting immune-related genes expressed by the cancer cells. To address that, the authors could test if there is low-level increased signals for the entire profile of immune cells (as expected by contaminating cells), or if this signals comes from very few immune-related genes (as expected by expression from the cancer cells). If the results of such analysis are inconclusive then the authors should be careful in how they report this effect and acknowledge the potential confounder.

4. The main text for this analysis is quite descriptive and could probably be optimized to better introduce the motivation and the results. Also, some discussion about how this relates to known differences in core/periphery in prostate and in other tumor types may be helpful. I would imagine that core-vs-periphery is one of the main questions that come up from spatial transcriptomics, but the authors seem to give little attention to this aspect even after doing the initial analysis.

Replies to the reviewers' comments

All changes are highlighted in purple color in the manuscript.

Reviewer #1

Comment (i)

The initial clustering shown by Supplementary. Fig. 6A is not reliable, as the tSNE plot does not indicate three distinct clusters. Of course, clustering algorithms will always produce some definition of clusters, but I would expect that a plot that demonstrates the clustering should show a pattern that supports the clustering, which is not the case in this figure. If the clustering is indeed robust, then perhaps the choice of presentation is suboptimal and it could be shown in another way (e.g. with a dendrogram). Otherwise, if the clustering is forced by the use of a clustering algorithm and the signal for this clustering is not robust then I would suggest using a different computational approach, such as PCA or NNMF, which may deal better with a continuous rather than discrete pattern, or to completely eliminate the clustering analysis.

Reply to Comment (i)

Indeed there is a challenge to track expression gradients by tSNE (or PCA) due to the intrinsic nature of gradients yet we believed that Suppl Fig 6A did visualize the gradual differences. Albeit, in the light of the reviewers' comment we have removed Figure 6A from the supplementary material. Instead, we investigated in more depth the gene expression adjacent to the cancer core across several tumors (Fig 4, Supplementary Figs. 17-18) as discussed in the replies below.

Comment (ii)

Are the results from the core/periphery analysis consistent between tumors? This analysis appears to include only one tumor, and while I understand that there is much more data for this compared to the other tumors, I do believe that it is important to make sure that there is some consistency with at least one other tumor, and perhaps report only the consistent effects. If the data for the other tumors is insufficient to perform a similar analysis, then perhaps the authors can just evaluate the effect - as defined in the first tumor – in the context of other tumors. However, a separate analysis in each tumor, followed by integration of the results across tumors, would be better.

Reply to Comment (ii)

In line with the comments from the reviewer, we have now performed similar core/periphery analysis on another tumor (sample 3.3, Gs 3+4) and compared the results to sample 1.2 (Gs 3+3) (Fig 4). We believe that the differences seen between the two samples stems from the fact that the cancer region in 1.2 is much smaller than in 3.3, which leads to different pathway activations. E.g. enrichment of metabolic pathways is observed in 1.2 (Oxidative phosphorylation, Pentose phosphate pathway, Citrate cycle); metabolic alteration is a hallmark of cancer. Yet, for sample 3.3, enrichment of Endocytosis, Lysosome and Phagosome is observed, which is reasonable since a large growing cancer causes abnormalities in tumor blood vessels. This leads to low oxygen supply and formation of necrotic areas with cell debris and dead cells. Interestingly, activation of the HIF-1 signaling pathway (activation is prompted by hypoxic conditions in order to control vascularization) is seen in the periphery of 3.3, which further speaks for low oxygen levels in 3.3. We observe some common

pathways between the tumors, e.g. pathways linked to cell proliferation (ECM-receptor interaction, PI3K-Akt signaling) and cell motility (regulation of the actin cytoskeleton, focal adhesion). We also investigated differences between the core and periphery of sample 2.4 (suspected cancer). Interestingly, the cancer area of sample 2.4 shows more similarities with the one in sample 1.2 (Fig. 4 and Supplementary Fig. 18) compared to sample 3.3. This is expected since both 1.2 and 2.4 display similar activities in the cancer factor.

Comment (iii)

Some of the signal in the periphery involves immune-related genes. It is difficult to conclude if these are due to contamination of these regions by few immune cells or if this is truly reflecting immune-related genes expressed by the cancer cells. To address that, the authors could test if there is low-level increased signals for the entire profile of immune cells (as expected by contaminating cells), or if this signals comes from very few immune-related genes (as expected by expression from the cancer cells). If the results of such analysis are inconclusive then the authors should be careful in how they report this effect and acknowledge the potential confounder.

Reply to Comment (iii)

We investigated in more detail the immune-related genes observed in the periphery. First, we asked a pathologist to re-annotate the region in 1.2 for immune cells (Supplementary Fig. 17a), and it demonstrated that there are regions surrounding the cancer containing immune cells. Furthermore, we investigated the genes responsible for the activated pathways in the periphery and identified high expression of NFKBIA (T-cell marker), IRF7 (Toll like marker) and HLA-C (regulator of immune system) (Supplementary Fig. 17a). Importantly, the periphery itself is not cancerous tissue. Hence, it is probably not the cancer cells that express immune-related genes but presence of immune cells in the surrounding tissue.

Comment (iii)

The main text for this analysis is quite descriptive and could probably be optimized to better introduce the motivation and the results. Also, some discussion about how this relates to known differences in core/periphery in prostate and in other tumor types may be helpful. I would imagine that core-vs-periphery is one of the main questions that come up from spatial transcriptomics, but the authors seem to give little attention to this aspect even after doing the initial analysis.

Reply to Comment (iii)

We agree that using the spatial transcriptomics techniques to investigate gene expression differences in core-vs-periphery is an important aspect. We have therefore re-organised the manuscript accordingly, so that some of the supplementary analysis are now included in the main manuscript and some of images are now part of Fig. 4 (Consequently, moved p63 staining to Supplementary Fig. 9) to give it more attention. In order to make it less descriptive, we also put more focus on describing the actual differences and similarities between the tumor samples.

REVIEWERS' COMMENTS:

Reviewer #1 (Remarks to the Author):

I am satisfied with the revisions and support the publication of this work